Comprehensive analyses of genetic diversities and population structure of the Guizhou Dong group based on 44 Y-markers

Zhang Hongling 229598103@qq.com
Huang Xiaolan
Jin Xiaoye
Ren Zheng
Wang Qiyan
Yang Meiqing
Xu Ronglan
Yuan Xiang
Yang Daiquan
Liu Hongyan
Shen Wanyi
Zhang Huiying
Que Yangjie
Huang Jiang mmm_hj@126.com
Department of Forensic Medicine, Guizhou Medical University , Guiyang, Guizhou , China
Azevedo Luisa
Electronic publication date: 2023 Sep 25
Publication date: 2023
Volume: 11
Electronic Location ID: e16183
Received 2023 May 24; Accepted 2023 Sep 5
Copyright: © 2023 Zhang et al.
Copyright year: 2023
Copyright holder: Zhang et al.
License: This is an open access article distributed under the terms of the Creative Commons Attribution License, which permits unrestricted use, distribution, reproduction and adaptation in any medium and for any purpose provided that it is properly attributed. For attribution, the original author(s), title, publication source (PeerJ) and either DOI or URL of the article must be cited.
License URL: https://creativecommons.org/licenses/by/4.0/

Keywords: Guizhou Dong, Forensic features, Population genetics, STR, Y-chromosome

Funding: Guizhou Province Education Department, Characteristic Region Project, Qian Education (2021) 065 Guizhou Innovation training program (202110660018) Guizhou Provincial Science and Technology (ZK(2022) General 355) Guizhou Education Department Young Scientific and Technical (2022) 215 Guizhou Scientific (2021) General 448 Qian Science Platform Talents (2020) 6012 Qian Science Support (2019) 2825 National Natural Science Foundation (No. 82160324) Guizhou Science Project, Qian Science Foundation (2020) 1Y353 This study was supported by the Guizhou Province Education Department, Characteristic Region Project, Qian Education KY No. (2021) 065; the Guizhou Innovation training program for college students (202110660018); the Guizhou Provincial Science and Technology Projects (ZK[2022) General 355 the Guizhou Education Department Young Scientific and Technical Talents Project, Qian Education KY NO. (2022) 215; the Guizhou Scientific Support Project, Qian Science Support (2021) General 448; the Guizhou “Hundred” High-level Innovative Talent Project, Qian Science Platform Talents (2020) 6012; the Guizhou Scientific Support Project, Qian Science Support (2019) 2825; the National Natural Science Foundation (No. 82160324); and the Guizhou Science Project, Qian Science Foundation (2020) 1Y353. The funders had no role in study design, data collection and analysis, decision to publish, or preparation of the manuscript.

==============================
Background

The non-recombining region of the human Y chromosome (NRY) is a strictly paternally inherited genetic marker and the best material to trace the paternal lineages of populations. Y chromosomal short tandem repeat (Y-STR) is characterized by high polymorphism and paternal inheritance pattern, so it has been widely used in forensic medicine and population genetic research. This study aims to understand the genetic distribution of Y-STRs in the Guizhou Dong population, provide reference data for forensic application, and explore the phylogenetic relationships between the Guizhou Dong population and other comparison populations.

Methods

Based on the allele profile of 44 Y-markers in the Guizhou Dong group, we estimate their allele frequencies and haplotype frequencies. In addition, we also compare the forensic application efficiency of different Y-STR sets in the Guizhou Dong group. Finally, genetic relationships among Guizhou Dong and other reference populations are dissected by the multi-dimensional scaling and the phylogenetic tree.

Results

A total of 393 alleles are observed in 312 Guizhou Dong individuals for these Y-markers, with allele frequencies ranging from 0.0032 to 0.9679. The haplotype diversity and discriminatory capacity for these Y-markers in the Guizhou Dong population are 0.99984 and 0.97440, respectively. The population genetic analyses of the Guizhou Dong group and other reference populations show that the Guizhou Dong group has the closest genetic relationship with the Hunan Dong population, and followed by the Guizhou Tujia population.

Conclusions

In conclusion, these 44 Y-markers can be used as an effective tool for male differentiation in the Guizhou Dong group. The haplotype data in this study not only enrich the Y-STR data of different ethnic groups in China, but also have important significance for population genetics and forensic research.

Introduction

The non-recombining region of the human Y chromosome (NRY) strictly follows a paternally inherited pattern, implying that it can only be transmitted from father to male descendants. Therefore, genetic markers on the NRY can be viewed as important markers to trace the paternal lineages of populations (Sofie et al., 2019). Since NRY is more susceptible to genetic drift caused by its small effective population size than other autosomal genetic markers (Jobling & Tyler-Smith, 2003), it usually shows diverse genetic distributions among different geographical populations, which makes the Y chromosome a powerful tool for investigating demographic events in human genetic history. Until now, the NRY has been widely applied in the studies of forensics, genealogical reconstruction, human origin, evolution and migration, population history, and medical genetics (Sofie et al., 2019; Jobling & Tyler-Smith, 2003; King & Jobling, 2009; Jobling & Tyler-Smith, 1995). For the NRY, Y chromosomal short tandem repeats (Y-STRs) are commonly used in forensic genetics and population genetics due to their high genetic polymorphism and paternal inheritance pattern. Since Y-STRs have an average mutation rate of 3.35 × 10−3, they can also be used to study the genetic composition and consanguinity issues of various populations (Ballantyne et al., 2010). Currently, many commercial Y-STR kits have been developed for forensic research. For example, the Power Plex® Y23 System, which is a five-dye system, combines 17 Y-STR loci in the currently commercially available Y-STR kit (Y-filer) with six new high-discrimination Y-STR loci (Thompson et al., 2013). In 2017, He et al. analyzed the genetic polymorphisms of these 23 Y-STRs in the Yi population of Sichuan Province in China. Obtained results revealed that these Y-STRs displayed relatively high genetic diversities and could be used to differentiate males in the Yi population (He et al., 2017). However, as the extant DNA database gathers several individuals, the match chance of the allelic profile of random individuals will increase if only these 23 Y-STRs are tested. Accordingly, more Y-markers should be added to the existing panels to improve their haplotype resolution power. Compared to the Power Plex Y23 system, the Goldeneye DNA Identification System Y Plus Kit (Beijing PeopleSpot Inc.) can simultaneously amplify 41 Y-STRs and 3 Y chromosomal Insertion/Deletion polymorphisms (Y-InDels) in a single well using the six fluorescent dye labeling technology. The kit combines high and low mutation rate loci, especially for three Y-InDels, which can be used as the slow mutation loci for familial search. Besides, they could also enhance the discriminatory capacity of the kit to some degree. We postulate that the panel could achieve higher male differentiation than the Power Plex Y23 system. Therefore, the forensic application value of the panel is further evaluated in the current study.

There are more than 40 ethnic minorities in the Guizhou province. The Dong ethnic group, also known as the Kam population, is one of the major ethnic groups living in Guizhou. It is generally believed that the modern Dong population developed from the ancient Baiyue tribe, which migrated to the Guizhou in the Han Dynasty (Wu, 1993). The Dong group has their own language that belongs to a branch of the Tai-Kadai language family. According to the sixth national census statistics, there are 1,431,900 Dong population in Guizhou, accounting for 4.12% of the province’s population and 11.41% of the minority population in Guizhou (Ma, 2010). These Dongs are mainly distributed in southeastern Guizhou Province and adjacent to Guangxi Zhuang Autonomous Region and Hunan Province (https://www.britannica.com/topic/Dong). So far, some genetic data on the Y-STRs of Hunan Dong (Feng et al., 2020), the autosomal STRs of Guangxi Dong (Guo et al., 2017a) and the mitochondrial DNA of Guizhou Dong (Ren et al., 2022) have been reported. However, there are no studies on the Y-STRs of the Dong population in Guizhou. Therefore, we investigate the genetic distributions of 44 Y-markers in the Guizhou Dong group based on the Goldeneye DNA identification system Y Plus kit. Furthermore, Chinese different populations collected in the Y chromosome haplotype reference database (YHRD, https://yhrd.org/) are used as reference populations to further explore the genetic background of the Guizhou Dong group. The study is expected to enrich the genetic data of the Guizhou Dong group, which could be viewed as the reference data set for forensic research in the Guizhou Dong group.

Materials and Methods

Sample information

In this study, bloodstain samples were collected from 312 unrelated healthy Dong individuals with written informed consent. All participants have lived in the Guizhou Province for at least three generations. This research is performed according to the guidelines of the Ethics Commission of Guizhou Medical University. The ethical permission is issued by the Ethics Committee of Guizhou Medical University (Approval Number: No. 224).

PCR amplification and capillary electrophoresis

According to the instructions of the Goldeneye DNA Identification System Y Plus kit, PCR amplification is performed on the aforementioned samples by using the 9700 PCR instrument (Applied Biosystems, Foster City, CA, USA). The total volume of the PCR reaction system is 10 μL, including 4× premix VII 2.5 μL, 5× Y Plus primer mixture 2.0 μL, deionized water 5.5 μL and 1.2 mm bloodstain. The reaction condition is initial denaturation at 95 °C for 2 min; then 27 cycles of 94 °C for 5 s, 60 °C for 90 s, 62 °C for 1 min; and then extension for 5 min at 60 °C, followed by a final hold at 15 °C. Next, we mix 1 μL amplified product with 10 μL deionized formamide and 0.4 μL ORG 500 internal standard; and then the cocktail is separated and detected on the ABI3500 Genetic Analyzer. Finally, allele nomenclature is conducted using GeneMapperID-X1.3 software (Thermo Fisher Scientific, Waltham, MA, USA) by comparing it to the kit’s allelic ladder.

Statistical analysis

Allele frequencies and haplotype frequencies of 41 Y-STRs and three Y-InDels in the Guizhou Dong population are calculated by the direct counting method. Gene diversity (GD) and haplotype diversity (HD) are calculated according to the formula ‘GD or HD = n (1-∑Pi 2)/(n-1)’, where Pi is the frequency of the i-th allele or haplotype and n is the total number of samples (Nei, 1973). Haplotype match probability (HMP) and discrimination capacity (DC) values are calculated according to the formulas: HMP = ∑Pi2 (Pi is the frequency of the haplotype), and DC = Ndiff/N (Ndiff and N mean the number of different haplotypes and the sample size, respectively). Population genetic analyses of Guizhou Dong and 16 reference populations are conducted by the following methods. Firstly, analysis of molecular variance (AMOVA) is used to estimate genetic distances (RST) between the Guizhou Dong group and other populations (Guizhou Bouyei (Luo et al., 2019), Yunnan Dai, Hunan Dong (Shu et al., 2015), Guizhou Han (Sun et al., 2019; Zhou et al., 2016), Yunnan Han (Kwak et al., 2005; Yanmei et al., 2010; Yin et al., 2022, 2020), Yunnan Hui (Xie et al., 2019), Guizhou Miao (Tang et al., 2020; Luo et al., 2021), Hulunbuir Mongolian (Wang et al., 2019), Sichuan Qiang, Qinghai Salar (Zhu et al., 2007), Qinghai Tibetan (Zhu et al., 2008; Cao et al., 2018), Guizhou Tujia, Guizhou Yi (Song et al., 2021), Yunnan Yi (Zhu et al., 2006; Fan et al., 2019), Gansu Yugur and Guangxi Zhuang (Wang et al., 2022; Guo et al., 2017b) by the online tool YHRD. Next, the phylogenetic tree of these populations is constructed by the MEGA v7.0 software (Kumar, Stecher & Tamura, 2016) with the method of Neighbor-Joining based on their pairwise RST values. In addition, the multi-dimensional scaling (MDS) of the Dong population and other reference populations is conducted by the SPSS v18.0 software (https://www.ibm.com/products/spss-statistics).

Results

Haplotype distributions, allele frequencies, and genetic diversities of 44 Y-markers

The different haplotypes and corresponding haplotype frequencies in the Guizhou Dong population are arranged in Table S1. A total of 304 haplotypes are observed in 312 individuals, 296 of which are unique. Allele distributions and GD values of 44 Y-markers in the Dong population of Guizhou are shown in Fig. 1 and Table S2. Two to 46 different alleles are observed per locus, resulting in a total of 393 alleles in 312 Guizhou Dong individuals, with allele frequencies ranging from 0.0032 to 0.9679. The locus with the highest GD value is DYS385 (0.9399), while the locus with the lowest GD value is DYS645 (0.0627). In addition, we find that nine Y-markers (DYS391, DYS388, DYS437, DYS438, DYS596, DYS645, rs199815934, rs759551978 and rs771783753) display relatively low GD values (<0.5) in the studied population. The other loci show relatively high GD values (>0.5) in the Guizhou Dong population, especially for DYS385, DYS527 and DYF387S1 loci whose GD values are higher than 0.9. For four multi-copy Y-STRs, we observe that these four loci (DYS385, DYF387S1, DYS527, and DYF404S1) have higher GD values than most single-copy Y-STRs, with GD values of 0.9399, 0.9363, 0.9170, and 0.8806, respectively. Microvariant alleles are also found at the DYS458 (14.1), DYS518 (37.2), DYS576 (18.1, 19.1 and 20.1) and DYS557 (18.3 and 17.3) loci. Bi-alleles are observed at the single-copy locus DYS557 (13,14), and tri-alleles are detected at two double-copy loci DYS527 (19,21,22) and DYF404S1 (13,14,16) (Table S1).

Figure 1 The numbers of alleles and gene diversities of 44 Y-markers in the Guizhou Dong population.

The blue line on behalf of GD values, and the yellow bar on behalf of alleles counts. Y axis on the left indicates allele count, and Y axis on the right indicates GD values.

Comparison of forensic efficacy of different Y sets in the Guizhou Dong population

The number of loci included by different Y kits and the number of overlapping loci between these kits are shown in Table S3. The DC, HD, HMP values and haplotype numbers for different Y-marker sets in the Guizhou Dong population are shown in Fig. 2 and Table S4. The HMP, HD and DC values of the 44 Y-markers are 0.00337, 0.99984 and 0.97440, respectively. The DC value increases from 0.72115 for 9 Y-STRs of the minimal set to 0.97440 for 44 Y-markers (the kit in this study). The HD value increases from 0.99143 for 9 Y-STRs to 0.99984 for 44 Y-markers. Furthermore, in Fig. 2, we find that with the increase of the number of Y-markers, DC and HD values gradually increase, while HMP gradually decreases. More importantly, the Goldeneye DNA identification system Y Plus kit in this study has higher DC values and higher HD values than other kits except for the Y filer Plus kit.

Figure 2 Forensic efficiency comparisons of different Y-STR sets in the Guizhou Dong population.

(A) Haplotype numbers and DC values of different Y-STRs kits in Guizhou Dong population. Y axis on the left indicates haplotype numbers, and Y axis on the right indicates discriminatory capacity. (B) HMP and HD values of different Y-STRs kits in Guizhou Dong population. Y axis on the left indicates haplotype match probability, and Y axis on the right indicates haplotype diversity.

Genetic relationships between the Guizhou Dong population and other reference populations

Firstly, the genetic relationships between the Dong population in Guizhou and 16 other reported populations are evaluated using the RST values, which represent genetic distances between populations. Table S5 summarizes the pairwise RST values in the Dong population of Guizhou and 16 other reference populations from China. Heatmap of the RST values is revealed in Fig. 3. The genetic distance between the Dong population in Guizhou and the Yi population in Yunnan is the largest (RST = 0.1573), followed by the Tibetan population in Qinghai (RST = 0.1550). This indicates that the genetic distances between the Dong population in Guizhou and the above two groups are relatively far. Genetically close to the Dong population in Guizhou are the Hunan Dong population (RST = 0.0162), Guizhou Tujia population (RST = 0.0211), Guizhou Yi population (RST = 0.0261) and Guizhou Han population (RST = 0.0283).

Figure 3 The heatmap based on the paired-Rst values for Guizhou Dong population and other reference populations.

The squares of different colors on the right represent different language families, and populations of the same language family show the same color. Besides, the larger area of the color blocks in the circular chart indicates a more distant genetic distance.

Next, based on the paired RST values, we conducted the MDS of the Dong group in Guizhou and other reference groups, as shown in Fig. 4A. The results show that the 17 groups can be roughly divided into three parts: the Qinghai Tibetan is in the upper right corner; the Dai and Yi nationalities of Yunnan are located directly below; the remaining groups are on the left. As for the Dong ethnic group studied, we find that this group is close to the Dong ethnic group in Hunan and the Tujia ethnic group in Guizhou. In addition, a phylogenetic tree is constructed using the Neighbor-Joining method based on the paired RST values of the 17 populations, as shown in Fig. 4B. The results show that the studied Dong ethnic group first gathers in a branch with the Hunan Dong ethnic group, and then with the Guizhou Tujia ethnic group, indicating that these groups have relatively close genetic relationships. These findings are consistent with the above MDS results.

Figure 4 Population genetic analyses of Guizhou Dong population and other reference populations.

(A) The MDS of Guizhou Dong and other populations; (B) the phylogenetic tree of Guizhou Dong and other populations.

Discussion

The results of this study indicate that the 44 Y-markers can be used for forensic applications and population genetic studies in the Dong population of Guizhou. Most of these 44 Y-markers display relatively high genetic polymorphisms (GD > 0.5) in the Guizhou Dong population. Even so, we find that DYS438, DYS391, DYS437 and DYS645 loci exhibit relatively low genetic diversities in the studied population. Similar results are also observed in Hunan Zhuang and Guangxi Zhuang populations (Feng et al., 2020; Guo et al., 2017a). We infer that these four Y-STRs might be selected for forensic application in European populations because these loci exhibit relatively high GD values in European populations, especially for DYS391, DYS437 and DYS438 loci (Hallenberg et al., 2005; Pickrahn et al., 2016). In addition, we observe that the GD values of DYS645, rs759551978 and rs771783753 loci are lower than 0.2, indicating that these three loci possess poor genetic diversities in the Guizhou Dong. Accordingly, we need to select more highly polymorphic Y-STRs to replace these loci to obtain better haplotype resolution. At the same time, trialleles are found at two double-copy loci (DYS527 and DYF404S1) in the Guizhou Dong population. Similar results are found in individuals of the Shandong Han population, which may be related to the extra copies of these loci on the Y chromosome (Chen et al., 2022). This needs to be verified by the Sanger sequencing. For these 44 Y-markers, we find that they show higher DC values than other Y-marker sets. However, no obvious difference between 44 Y-markers and 27 Y-STRs (the Y filer Plus kit) is seen in the Guizhou Dong population, which may be related to the relatively small sample size (312) of the studied population. As the increasing of the sample size, we postulate that these 44 Y-markers will display higher DC and HD values than the Y filer Plus kit. Anyway, we think that these 44 Y-markers could be viewed as a valuable tool for distinguishing unrelated male individuals in the Guizhou Dong population based on their HD and DC values.

In Fig. 3, the genetic distance between Guizhou Dong and Hunan Dong is the closest, followed by the Guizhou Tujia population. In Figs. 4A and 4B, the Guizhou Dong group and Hunan Dong group are also clustered together, which indicates that the same population in different regions is genetically closer. Conversely, the genetic distance from the Guizhou Dong is the farthest from the Qinghai Tibetan population, followed by the Gansu Yugur, Mongolian and Yunnan Dai populations. The distant genetic relationship between the Dong population in Guizhou and the Tibetan population in Qinghai may be due to the fact that most Tibetans live at high altitudes and have little contact with other populations, which makes them genetically distant from other populations (Lu et al., 2016). In the previous study on 30 InDel loci of the Dong population, Liu et al. (2020) also found that the Dong population in Guizhou and the Tibetan population in Qinghai had high genetic differentiation. Nonetheless, the exploration of the genetic structure of the Dong population in Guizhou is not enough, and it is necessary to further study the genetic composition of the Dong population in Guizhou by using more genetic markers, such as whole genome sequencing, to clarify the genetic background of the Dong population in Guizhou.

Conclusion

We first assess the genetic distributions of 44 Y-markers in the Guizhou Dong population based on the Goldeneye DNA Identification System Y Plus kit. Among these loci, most markers exhibit high genetic polymorphisms in the Guizhou Dong population. The HD and DC values of 44 Y-markers show that the kit is capable of providing relatively high-resolution haplotypes, and these 44 loci can be well applied to forensic applications in the Dong population of Guizhou. At the same time, the haplotype data in this study can also provide valuable reference data for population genetics studies in the future.

Supplemental Information

Supplemental Information 1 Haplotype frequencies of 44 Y-markers of Guizhou Dong population (N = 312).

Click here for additional data file.

Supplemental Information 2 Allele frequencies and gene diversities of 44 Y-markers loci in Guizhou Dong population (n = 312).

Click here for additional data file.

Supplemental Information 3 Loci information of different Y sets.

Click here for additional data file.

Supplemental Information 4 Evaluation of the effectiveness of different Y-STR sets.

Click here for additional data file.

Supplemental Information 5 The pairwise RST values in the Dong population of Guizhou and 16 other reference populations from China.

Click here for additional data file.

We are grateful to all volunteers for providing their blood samples.

Additional Information and Declarations

Competing Interests

Author Contributions

Human Ethics

Data Availability

The authors declare that they have no competing interests.

Hongling Zhang conceived and designed the experiments, prepared figures and/or tables, authored or reviewed drafts of the article, and approved the final draft.

Xiaolan Huang conceived and designed the experiments, performed the experiments, analyzed the data, prepared figures and/or tables, authored or reviewed drafts of the article, and approved the final draft.

Xiaoye Jin analyzed the data, prepared figures and/or tables, and approved the final draft.

Zheng Ren analyzed the data, prepared figures and/or tables, and approved the final draft.

Qiyan Wang analyzed the data, prepared figures and/or tables, and approved the final draft.

Meiqing Yang analyzed the data, prepared figures and/or tables, and approved the final draft.

Ronglan Xu performed the experiments, prepared figures and/or tables, and approved the final draft.

Xiang Yuan performed the experiments, prepared figures and/or tables, and approved the final draft.

Daiquan Yang performed the experiments, authored or reviewed drafts of the article, and approved the final draft.

Hongyan Liu performed the experiments, authored or reviewed drafts of the article, and approved the final draft.

Wanyi Shen performed the experiments, authored or reviewed drafts of the article, and approved the final draft.

Huiying Zhang performed the experiments, authored or reviewed drafts of the article, and approved the final draft.

Yangjie Que performed the experiments, authored or reviewed drafts of the article, and approved the final draft.

Jiang Huang conceived and designed the experiments, prepared figures and/or tables, and approved the final draft.

The following information was supplied relating to ethical approvals (i.e., approving body and any reference numbers):

The Ethics Committee of Guizhou Medical University.

The following information was supplied regarding data availability:

The raw data is available in the Supplemental Files.

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
