# Peer review of "Comprehensive analyses of genetic diversities and population structure of the Guizhou Dong group based on 44 Y-markers"

_PeerJ, doi:10.7717/peerj.16183_

## Round 0.1 · original submission · Major Revisions

I have carefully considered your manuscript and although I find it scientifically sound, it needs significant revision. I invite you to carefully address all the issues provided by the two reviewers. Additionally, the clarity and precision of the language needs to be improved through editing.

Yours sincerely,

Reviewer 1 ·

Basic reporting

This article investigates the genetic distribution of 44 Y-markers in the Guizhou Dong population and explores the genetic relationship between the Guizhou Dong population and other reference populations. But some issues still need to be addressed. It may be more meaningful for future research if the following questions are well addressed.
1. Your manuscript needs careful editing and particular attention to English grammar, spelling, and sentence structure.And please consider careful language editing. In the Introduction,statements such as "NRY can be viewed as the important marker" are inappropriate, Moreover, the Y-STRs in "more Y-STRs should be added into the existing panels" should be consistent with the previous text, and it is more appropriate to change it to Y-markers.
2. I suggest that authors should provide ethic approval number in the main text.
3. Figure 3 and Figure 4 should give the actual values in supplementary table.
4. Please further introduce the benefit of the three Y-indels in the kit.
5. Authors should further describe haplotype distributions of these 44 Y-markers in the Guizhou Dong group.

Experimental design

See above

Validity of the findings

See above

Additional comments

None

Reviewer 2 ·

Basic reporting

1) Language
The English language should be improved to allow an easy understanding by an international audience. While the orthography is correct, the grammar should be improved. The article is lacking the correct application of plural and singular, contains colloquial language (“till now” in line 47 is a short-form of “until now” or “mainstream genetic markers” in lines 50-51), occasionally contains the wrong tempus (for statements that are a matter of fact, use present tense, e.g. line 159 “the Rst value, which presents genetic distances” and line 160 “Figure 3a shows”), or the wrong application of parts of speech (e.g. “Since NRY is more susceptibility to genetic drift”). Overall, the whole article needs to be proofread and reviewed to improve the overall readability.

2) Introduction
Lines 52-54: “Y-STRs have an average mutation rate of 0.2% per generation…” (Hayder et al.2020)
Mutation rates and mutation frequencies should not be used interchangeably. Mutation rates are given as number of mutations per DNA base pair per generation, mutation frequencies are usually given in percentage and describe the fraction of individuals within a population with a certain mutation. The cited paper by Hayder et al. describes an “average mutation frequency of 0.2% per generation”. In the same paper, it is not explained where this number comes from and if it is restricted to a specific population, like e.g. the Iraqi one. If you would like to use mutation frequencies, a more reliable and better explained source of Y-STR mutation frequencies can be found in “Advanced Topics in Forensic DNA Typing: Interpretation” by John Butler, which gives a slightly different average Y-STR mutation frequency. However, more commonly the community would refer to mutation rates (e.g. in “Mutability of Y-Chromosomal Microsatellites: Rates, Characteristics, Molecular Bases, and Forensic Implications” by Ballantyne et al. 2010), including the mutation rates of standard Y-STRs and rapidly mutating Y-STRs. I would recommend referring to mutation rates.

Line 58: “in the Yi people of the Sichuan Province [in China]”
You could mention in the introduction which country the population of interest is located in.

Lines 60-62: “However, more occasional matches may occur in the DNA database for these 23 Y-STRs as the extant DNA database gather a number of individuals.”
I am not sure if I understand this sentence correctly, but the more relevant Y-STRs are typed, the lower the chance to find a random match.

For some of the citations, the author names need to be adjusted, e.g. the first citation by Jobling and Tyler-Smith, or the fifth source.

Line 79: “However, there are few studies on Y-STRs of Dong population in Guizhou”
Citations missing

3) Main text

Line 131-132: “Totally, 2-46 alleles were observed in 312 Guizhou Dong individuals for these Y-markers”
Two to 46 different alleles were observed per locus, resulting in a total of 393 alleles in 312 Guizhou Dong individuals.

Line 136: You can omit “in spite of this”. Just because some loci do not have a high variance in your population, does not imply that other loci cannot be variable.

Figure 1: Please label your axes titles.

Lines 151-153: “Unsurprisingly, the lowest number of haplotypes and the lowest HD value were observed at the minimal haplotype, whereas the highest number of haplotypes were observed in the Goldeneye DNA Identification System Y Plus kit”
This sentence is not necessary.

Figure 2: Again I am missing labels at the axes. Also the y-axis to the right is not fitting for HD and HMP, since the reader cannot see any changes. You could transform the dataset or include these parameters in a different way. You could also include your own 44 Y-markers for a direct comparison (you can do this for Supplementary Table 3 as well). Please define “All”: which markers does this include? What the reader can take from this figure is, that the more loci are included in a panel, the higher the DC and HD, and the smaller the HMP. The interesting part would be to find out whether the right choice of markers has a bigger impact on the statistical power of a panel. For this, you could also include the number of loci per kit and which of these are overlapping between the kits, e.g. as a Supplementary Figure/Table. The manuscript would benefit from elaborating on the unique markers of the 44 marker set compared to the other panels, especially the Yfiler plus.

Figure 3a: You can label the color bar with "Rst". Is there a second layer to this heatmap in the pieces of the slices in the pie charts? There seems to be an error in your figure: Your main population Guizhou Dong is colored in yellow, which stand for the language family Hmong-Mien, but in your introduction you assign this population to Tai-Kadai. This is the same language family we see in Figure 4a as well. I think there are also 2 typographical errors: HunnanHan and HunnanHui instead of YunnanHan and YunnanHui.

Figure 3b could also be a normal bar charts, which would improve the readability. What does this figure add? It is a different representation of the first column of Figure 3a, where the reader can see that Qinghai Tibetan and Yunnan Yi have the largest Rst values when compared to Guizhou Dong. I think this figure can be omitted.

Line 160: “of 17 population pairs” sounds like there are 17 pairs in the end, which is not true. There are pairs between the 17 populations.

Lines 175-178: When describing closeness based on the phylogenetic tree in Figure 4b, the reader needs more information on how the phylogenetic tree was generated. This information would give clarity on whether the branch lengths are representative of genetic closeness. Further, one needs to be aware that circular phylogenetic trees tend to distort branch lengths. A better alternative would be the application of Networks, representing mutational step difference between haplotypes. Traditionally utilized softwares in population genetics would be the fluxus-engineering software. There you can also give different loci different weights, depending on their robustness in your population, e.g. loci that are highly variable in your population (e.g. DYS389) could be assigned a greater weight.

Lines 181-184: “Most of these 44 Y-markers were well polymorphic,but DYS438, DYS391, DYS437 and DYS645 loci were also observed as low diverse markers in four ethnic minorities in Hunan and Guangxi Zhuang populations(Feng et al. 2020; Guo et al. 2017b).”
This sentence needs to be changed to be more understandable. Further, an explanation for the lack of variation among the presented populations is that these markers most likely were selected based on European populations, as most of these are included in commercial Y-STR kits.

Lines 196-197: “The genetic relationship results presented in our study are consistent with each other and with many previous studies”
Please support this with articles.

Experimental design

Line 125: What kind of phylogenetic tree was constructed? I assume it is based on the 23/27/44 Y-STR data. Is it rooted or unrooted, what kind of algorithm was used for the tree generation within MEGA?

Figure 1: It could have been interesting to see which individuals vary in the loci that have a low GD. Are these samples from different region within Guizhou? Is there anything different about these individuals?

Figure 4b: As described above, more information on the rootedness and tree construction method should be given.

Validity of the findings

Line 138: “For four multi-copy Y-STRs” out of how many multi-copy STRs? What do you observe for other multi-copy loci?

You mention four multi-locus Y-STRs in lines 138-139 (DYS385, DYF387S1, DYS527, and DYF404S1) and later the following three loci: locus DYS557, DYS527 and DYF404S1(caused by duplication, lines 142-144). You could include loci DYS385 and DYF387S1 in the paragraph in lines 142-144 to make the explanations of multi-copy loci complete. If there are more multi-copy loci you could include them here as well.

Lines 160-162 “We found that Tibetan population in Qinghai and Yi population in Yunnan showed relatively high RST values with other populations”
The authors could give a Rst cut-off value for defining it as “relatively high”, compared to YunnanDai.

Lines 162-163: “populations from the same language group showed relatively low RST values” AND
Lines 201-202: “In addition, we found that populations from the same language family or common origin tend to have relatively low genetic differentiation.”
I am struggling to find results that support this statement (Figure 3a, 4a).

Lines 193-196: “Meanwhile, the kit we used showed high HD and DC values in Guizhou Dong people, which indicated these Y-STRs could be viewed as a valuable tool for male differentiation in the Guizhou Dong population.”
But Yfiler plus offered similar HD while having a smaller number of markers. This could be elaborated.

Lines 197-201: “in a previous study based on mitochondria DNA, Ren et al. found that the genetic relationship between Guizhou Dong group and Guizhou Miao group was the closest (Ren et al. 2022). Thus, we postulated that the neighboring geographical locations among Guizhou Dong and other groups in Guizhou might promote their gene exchange, which lead to their relatively close genetic affinities”
But these findings by Ren et al. are not supporting the findings of the authors’. GuizhouDong and GuizhouMiao do not have smaller Rst values compared to many of the other populations. Ren et al. are analyzing mitochondrial SNPs, while the authors here are analyzing Y-chromosomal STRs. Thus, you are comparing maternal pre-historic events with paternal historic events (due to the differing inheritance patterns and mutation rates).

Lines 202-204: ” Guo et al. also observed that the Dong people of Guangxi and Guizhou were clustered in the same branch through 17 autosomal STR (Guo et al. 2017b), which was similar to our results.”
In Guo et al. Guangxi and Guizhou clustered in a phylogenetic tree based on autosomal STRs, when compared to Yunnan populations, which are not covered in this study (except YunnanYi). In the current study Guangxi and Guizhou did not show a clear clustering (neither in the phylogenetic tree nor in the MDS plot).

Lines 209-211: “As most Tibetans live at high altitudes and have little contact with other populations, this makes them genetically distant from the rest of the population(Lu et al. 2016).”
Are the Tibetans the only populations that live isolated from the other populations? Maybe there is some literature providing information on the interaction between all the populations of interest. Isolated populations will experience more genetic drift and accumulate genetic variants different from the other populations.

Lines 220-221: “the kit could be regard as a valuable tool for forensic male identification”
Using the kit for male identification was not tested in this study. Due to the nature of (nearly) identical Y-STRs between close male relatives, the study would need to include male relatives. Then one could judge the ability to differentiate closely related individuals or the identification of males.

Additional comments

Thank you to the authors for addressing this interesting research question and your work. I hope you decide to address my suggestions, as I think it will improve the quality of the manuscript.

---

## Round 0.2 · Minor Revisions

Please address the issues of the reviewer and correct including the minor language errors.

Reviewer 2 ·

Basic reporting

Thank you to the authors, who did a good job addressing inconsistencies in the previous version of their manuscript. I think the manuscript has greatly improved, and the figures benefitted from the revision as well. I agree to most answers of the authors to my comments and to the revisions in the manuscript.
Although there are still a few language errors, overall the text is understandable.

I only have two small more concrete comments:

1) In your discussion section you can add literature supporting your statement: “We infer that these four Y-STRs might be selected for forensic application in European populations because these loci exhibit relatively high GD values in European populations, especially for DYS391, DYS437 and DYS438 loci”

2) Towards the end of your discussion you write “whole genome SNP sequencing”, either you mean whole genome sequencing or targeted sequencing.

Experimental design

no comment

Validity of the findings

no comment

---

## Round 0.3 · accepted · Accept

The authors have addressed all of the reviewer's comments.